# ONLINE META-LEARNING

## ABSTRACT

A central capability of intelligent systems is the ability to continuously build upon previous experiences to speed up and enhance learning of new tasks. Two distinct research paradigms have studied this question. Meta-learning views this problem as learning a prior over model parameters that is amenable for fast adaptation on a new task, but typically assumes the set of tasks are available together as a batch. In contrast, online (regret based) learning considers a sequential setting in which problems are revealed one after the other, but conventionally train only a single model without any task-specific adaptation. This work introduces an online meta-learning setting, which merges ideas from both the aforementioned paradigms to better capture the spirit and practice of continual lifelong learning. We propose the *follow the meta leader* algorithm which extends the MAML algorithm to this setting. Theoretically, this work provides an $\mathcal{O}(\log T)$ regret guarantee with one additional higher order smoothness assumption (in comparison to the standard online setting). Our experimental evaluation on three different large-scale tasks suggest that the proposed algorithm significantly outperforms alternatives based on traditional online learning approaches.

## 1 INTRODUCTION

Two distinct research paradigms have studied how prior tasks or experiences can be used by an agent to inform future learning. Meta-learning (Schmidhuber, 1987) casts this as the problem of *learning to learn*, where past experience is used to acquire a prior over model parameters or a learning procedure. Such an approach, where we draw upon related past tasks and form associated priors, is particularly crucial to effectively learn when data is scarce or expensive for each task. However, meta-learning typically studies a setting where a set of meta-training tasks are made available together upfront as a batch. In contrast, online learning (Hannan, 1957) considers a sequential setting where tasks are revealed one after another, but aims to attain zero-shot generalization without any task-specific adaptation. We argue that neither setting is ideal for studying continual lifelong learning. Meta-learning deals with learning to learn, but neglects the sequential and non-stationary nature of the world. Online learning offers an appealing theoretical framework, but does not generally consider how past experience can accelerate adaptation to a new task. In this work, we motivate and present the *online meta-learning* problem setting, where the agent simultaneously uses past experiences in a sequential setting to learn good priors, and also adapt quickly to the current task at hand.

**Our contributions:** In this work, we first formulate the online meta-learning problem setting. Subsequently, we present the *follow the meta-leader (FTML)* algorithm which extends MAML (Finn et al., 2017) to this setting. FTML is analogous to follow the leader in online learning. We analyze FTML and show that it enjoys a $\mathcal{O}(\log T)$ regret guarantee when competing with the best meta-learner in hindsight. In this endeavor, we also provide the first set of results (under any assumptions) where MAML-like objective functions can be provably and efficiently optimized. We also develop a practical form of FTML that can be used effectively with deep neural networks on large scale tasks, and show that it significantly outperforms prior methods in terms of learning efficiency on vision-based sequential learning problems with the MNIST, CIFAR, and PASCAL 3D+ datasets.

### 1.1 THE ONLINE META-LEARNING SETTING

Due to space constraints, we review the foundations of our work (meta-learning and online learning) in Appendix A. We consider a general sequential setting where an agent faces tasks one after another. Each task corresponds to a *round*, denoted by $t$. In each round, the goal of the learner is to determine model parameters $\mathbf{w}_t$ that perform well for the corresponding task at that round. This is monitored by $f_t : \mathbf{w} \in \mathcal{W} \to \mathbb{R}$, which we would like to be minimized. Crucially, we consider a setting where the agent can perform some local *task-specific* updates to the model before it is deployed and evaluated

in each round. This is realized through an update procedure, which at every round $t$ is a mapping $\boldsymbol{U}_t : \mathbf{w} \in \mathcal{W} \to \tilde{\mathbf{w}} \in \mathcal{W}$. This procedure takes as input $\mathbf{w}$ and returns $\tilde{\mathbf{w}}$ that performs better on $f_t$. One example for $\boldsymbol{U}_t$ is a step of gradient descent (Finn et al., 2017): $\boldsymbol{U}_t(\mathbf{w}) = \mathbf{w} - \alpha\nabla\hat{f}_t(\mathbf{w})$. Here, $\nabla\hat{f}_t$ is potentially an approximate gradient of $f_t$, as for example obtained using a mini-batch of data from the task at round $t$. The overall protocol for the setting is as follows:

1. At round $t$, the agent chooses a model defined by $\mathbf{w}_t$.
2. The world simultaneously chooses task defined by $f_t$.
3. The agent obtains access to $\boldsymbol{U}_t$, and uses it to update parameters as $\tilde{\mathbf{w}}_t = \boldsymbol{U}_t(\mathbf{w}_t)$.
4. The agent incurs loss $f_t(\tilde{\mathbf{w}}_t)$. Advance to round $t+1$.

The goal for the agent is to minimize regret over the rounds. Recall that regret is the difference in the cumulative loss incurred by the learning agent compared to the best offline algorithm within a comparator class (see Appendix A for a review of online learning and regret). A highly ambitious comparator is the best meta-learned model in hindsight. Let $\{\mathbf{w}_t\}_{t=1}^{T}$ be the sequence of models generated by the algorithm. Then, the regret we consider is:

$$\text{Regret}_T = \sum_{t=1}^{T} f_t\big(\boldsymbol{U}_t(\mathbf{w}_t)\big) - \min_{\mathbf{w}} \sum_{t=1}^{T} f_t\big(\boldsymbol{U}_t(\mathbf{w})\big). \tag{1}$$

Notice that we allow the comparator to adapt locally to each task at hand; thus the comparator has strictly more capabilities than the learning agent, since it is presented with all the task functions in batch mode. Achieving sublinear regret suggests that the agent is improving over time and is competitive with the best meta-learner in hindsight (since $\text{Regret}_T/T \to 0$ as $T \to \infty$). In the batch setting, meta-learning has been observed to perform better than jointly training a single model to work on all the tasks (Santoro et al., 2016; Finn et al., 2017; Vinyals et al., 2016; Li & Malik, 2017; Ravi & Larochelle, 2017). Thus, we may hope that learning sequentially, but still being competitive with the best meta-learner in hindsight, provides a significant leap in continual learning.

## 2 ALGORITHM AND ANALYSIS

Our algorithmic approach, *follow the meta leader* (FTML), takes inspiration from follow the leader (FTL) (Hannan, 1957; Kalai & Vempala, 2005) and adapts it to the online meta learning setting. FTML chooses the parameters according to:

$$\mathbf{w}_{t+1} = \arg\min_{\mathbf{w}} \tilde{F}_t(\mathbf{w}), \quad \text{where} \quad \tilde{F}_t(\mathbf{w}) \overset{\text{def}}{:=} \sum_{k=1}^{t} f_k\big(\boldsymbol{U}_k(\mathbf{w})\big). \tag{2}$$

This can be interpreted as the agent playing the best meta-learner in hindsight if the learning process were to stop at round $t$. In practice, we may not have full access to $f_k(\cdot)$, such as when it is the population risk and we only have a finite size dataset. In such cases, we will draw upon stochastic approximation algorithms to solve the optimization problem in Eq. (2).

We concentrate on the case where the update procedure is 1 step of stochastic gradient descent, as in the case of MAML, i.e. $\boldsymbol{U}_t(\mathbf{w}) = \mathbf{w} - \alpha\nabla\hat{f}_i(\mathbf{w})$. We assume that each loss function, $\{f_t, \hat{f}_t\}$ $\forall t$, is $C^2-$smooth (i.e. $G-$Lipschitz, $\beta-$smooth, and $\rho-$Lipschitz Hessian) and $\mu-$strongly convex. See Appendix B for more details and implications of these assumptions, and connections to standard online learning setting. Importantly, these assumptions *do not* trivialize the meta-learning setting. There is a clear separation between meta-learning and joint training even for linear regression (simplest strongly convex problem). See Appendix E for an example illustration.

**Theorem 1.** *Suppose $f$ and $\hat{f} : \mathbb{R}^d \to \mathbb{R}$ satisfy the stated assumptions. Let $\tilde{f}$ be the function evaluated after the gradient update procedure, i.e. $\tilde{f}(\mathbf{w}) := f\big(\mathbf{w} - \alpha\nabla\hat{f}(\mathbf{w})\big)$. If the step size is selected as $\alpha \leq \min\{\frac{1}{2\beta}, \frac{\mu}{8\rho G}\}$, then $\tilde{f}$ is $\tilde{\beta} = 9\beta/8$ smooth and $\tilde{\mu} = \mu/8$ strongly convex.*

Since the objective function is convex, we may expect first-order optimization methods to be effective, since gradients can be efficiently computed with standard automatic differentiation libraries (as discussed in Finn et al. (2017)). In fact, this work provides the first set of results (under any assumptions) under which MAML-like objective function can be provably and efficiently optimized. An immediate corollary of our main theorem is that FTML now enjoys the same regret guarantees (up to constant factors) as FTL does in the comparable setting (with strongly convex losses).

**Corollary 1.** *(inherited regret bound for FTML) Suppose that for all $t$, $f_t$ and $\hat{f}_t$ satisfy assumptions 1 and 2. Suppose that the update procedure in FTML (Eq. 2) is chosen as $U_t(\mathbf{w}) = \mathbf{w} - \alpha\nabla\hat{f}_t(\mathbf{w})$ with $\alpha \leq \min\{\frac{1}{2\beta}, \frac{\mu}{8\rho G}\}$. Then, FTML enjoys the following regret guarantee*

$$\sum_{t=1}^{T} f_t\big(U_t(\mathbf{w}_t)\big) - \min_{\mathbf{w}} \sum_{t=1}^{T} f_t\big(U_t(\mathbf{w})\big) = \mathcal{O}\left(\frac{32G^2}{\mu}\log T\right)$$

More generally, our main theorem implies that there exists a large family of online meta-learning algorithms that enjoy sub-linear regret, based on the inherited smoothness and strong convexity of $\tilde{f}(\cdot)$. See Hazan (2016); Shalev-Shwartz (2012); Shalev-Shwartz & Kakade (2008) for algorithmic templates to derive sub-linear regret based algorithms.

## 2.1 Practical Algorithm

In the previous section, we derived a theoretically principled algorithm for convex losses. However, many problems of interest in machine learning and deep learning have a non-convex loss landscape, where theoretical analysis is challenging. Nevertheless, algorithms originally developed for convex losses like gradient descent and AdaGrad Duchi et al. (2011) have shown promising results in practical non-convex settings. The main considerations for developing a practical variant are: (a) the optimization problem in Eq. (2) has no closed form solution, and (b) we do not have access to the population risk $f_t$ but only a subset of the data. To overcome both these limitations, we can use iterative stochastic optimization algorithms. Specifically, by adapting the MAML algorithm of Finn et al. (2017), we can use stochastic gradient descent with a minibatch of data, $\mathcal{D}_k^{\text{tr}}$, as the update rule. Similarly, SGD with an independently-sampled minibatch of data, $\mathcal{D}_k^{\text{val}}$, can be used to optimize the parameters. The gradient computation is described below:

$$\mathbf{g}_t(\mathbf{w}) = \nabla_{\mathbf{w}} \,\mathbb{E}_{k\sim\nu^t}\mathcal{L}\big(\mathcal{D}_k^{\text{val}}, U_k(\mathbf{w})\big), \;\; \text{where} \;\; U_k(\mathbf{w}) \equiv \mathbf{w} - \alpha\,\nabla_{\mathbf{w}}\,\mathcal{L}\big(\mathcal{D}_k^{\text{tr}}, \mathbf{w}\big) \tag{3}$$

Here, $\nu^t(\cdot)$ denotes a sampling distribution for the previously seen tasks (we use a uniform distribution in our experiments). $\mathcal{L}(\mathcal{D}, \mathbf{w})$ is the loss function (e.g. cross-entropy) averaged over the datapoints $(\mathbf{x}, \mathbf{y}) \in \mathcal{D}$ for the model with parameters $\mathbf{w}$. While $U_t$ in Eq. (3) includes only one gradient step, we observed that it is beneficial to take multiple gradient steps in the inner loop (i.e., in $U_t$), which is consistent with prior works (Finn et al., 2017; Grant et al., 2018; Antoniou et al., 2018).

The overall algorithmic procedure proceeds as follows. We first initialize a task buffer $\mathcal{B} = [\,]$. When presented with a new task at round $t$, we add task $\mathcal{T}_t$ to $\mathcal{B}$ and initialize a task-specific dataset $\mathcal{D}_t = [\,]$, which is appended to as data incrementally arrives for task $\mathcal{T}_t$. As new data arrives for task $\mathcal{T}_t$, we iteratively compute and apply the gradient in Eq. (3), which uses data from all tasks seen so far. Once all of the data (finite-size) has arrived for $\mathcal{T}_t$, we move on to task $\mathcal{T}_{t+1}$.

## 3 Experiments

Our experimental evaluation studies the practical FTML algorithm (Section 2.1) in the context of vision-based online learning problems. These problems include synthetic modifications of the MNIST dataset, pose detection with synthetic images based on PASCAL3D+ models (Xiang et al., 2014), and realistic online image classification experiments with the CIFAR-100 dataset. The aim of our experimental evaluation is to study the following questions: (1) can online meta-learning (and specifically FTML) be successfully applied to multiple non-stationary learning problems? and (2) does online meta-learning (FTML) provide empirical benefits over prior methods?

To this end, we compare to the following algorithms: (a) Train on everything (TOE) trains on all available data so far (including $\mathcal{D}_t$ at round $t$) and trains a single predictive model. This model is directly tested without any specific adaptation since it has already been trained on $\mathcal{D}_t$. (b) Train from scratch, which initializes $\mathbf{w}_t$ randomly, and finetunes it using $\mathcal{D}_t$. (c) Joint training with fine-tuning, which at round $t$, trains on all the data jointly till round $t-1$, and then finetunes it specifically to round $t$ using only $\mathcal{D}_t$. This corresponds to the standard online learning approach where FTL is used (without any meta-learning objective), followed by task-specific fine-tuning. We compare the algorithms on two metrics: (1) task performance (e.g. classification accuracy) for each task in the sequence; and (2) learning efficiency or amount of data needed to reach a proficiency threshold (e.g. 90% classification accuracy).

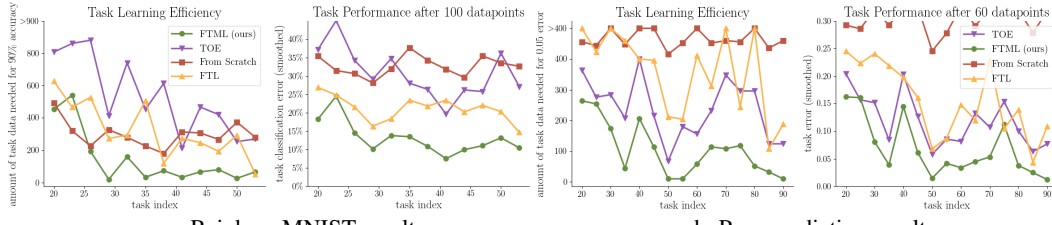

a. Rainbow MNIST results         b. Pose prediction results

Figure 1: Amount of data needed to learn each new task and task performance after a limited number of datapoints on the current task for each domain. Lower is better for all plots. FTML can learn new tasks more and more efficiently as each new task is received, demonstrating effective forward transfer.

We note that TOE is a very strong point of comparison, capable of reusing representations across tasks, as has been proposed in a number of prior continual learning works (Rusu et al., 2016; Aljundi et al., 2017; Wang et al., 2017). However, unlike FTML, TOE does not explicitly learn the structure across tasks. Thus, it may not be able to fully utilize the information present in the data, and will likely not be able to learn new tasks with only a few examples. Further, the model might incur negative transfer if the new task differs substantially from previously seen ones, as has been observed in prior work (Parisotto et al., 2016). FTL with fine-tuning represents a natural online learning comparison, which in principle should combine the best parts of learning from scratch and TOE, since this approach adapts specifically to each task *and* benefits from prior data. However, in contrast to FTML, this method does not explicitly meta-learn and hence may not fully utilize any structure in the tasks, and may also overfit in the fine-tuning stage.

In the rainbow MNIST experiment, we transform the digits in a number of ways to create different tasks, such as 7 different colored backgrounds, 2 scales (half size and original size), and 4 rotations of 90 degree intervals. A task involves correctly classifying digits with a randomly sampled background, scale, and rotation. As seen in the results curves in Figure 1a, FTML learns tasks more and more quickly with each new task. We also observe that FTML substantially outperforms the baselines in both efficiency and end performance. FTL is better than TOE since it performs task-specific adaptation, but still worse than FTML. We hypothesize that, while TOE and FTL improve in efficiency over the course of learning as they see more tasks, they struggle to prevent negative transfer on each new task. Our last observation is that training independent models does not learn efficiently, compared to models that incorporate data from other tasks; but, their asymptotic performance with a large data size is similar.

Our next experiment studies a 3D pose prediction problem. Each task involves learning to predict the global position and orientation of an object in an image. We construct a dataset of synthetic images using 50 object models from 9 different object classes in the PASCAL3D+ dataset (Xiang et al., 2014), rendering the objects on a table using the renderer accompanying the MuJoCo physics engine (Todorov et al., 2012). To place an object on the table, we select a random 2D location, as well as a random azimuthal angle. Each task corresponds to a different object with a randomly sampled camera angle. For the loss functions, we use mean-squared error, and set the proficiency threshold to an error of 0.05. We show the results of this experiment in Figure 1b. The results demonstrate that meta-learning can significantly improve both efficiency and performance of new tasks over the course of learning, solving many of the tasks with only 10 datapoints. Unlike the previous settings, TOE substantially outperforms the independent task models, indicating that it can effectively make use of the previous data from other tasks (likely due to greater structural similarity in this task). However, the efficiency and performance of online meta-learning demonstrates that even better transfer can be accomplished by explicitly optimizing for the ability to quickly and effectively learn new tasks. Appendix C presents a more detailed evaluation and results on the CIFAR task.

## 4 CONCLUSION

In this paper, we introduced the online meta-learning problem statement, with the aim of connecting the fields of meta-learning and online learning. Online meta-learning provides, in some sense, a more natural perspective on the ideal real-world learning procedure. An intelligent agent interacting with a constantly changing environment should utilize streaming experience to both master the task at hand, and become more proficient at learning new tasks in the future. We summarize prior work related to our setting in Appendix D. For the online meta-learning setting, we proposed the FTML algorithm and showed that it enjoys logarithmic regret. We then illustrated how FTML can be adapted to a practical algorithm. Our experimental evaluations demonstrated that the proposed practical variant outperforms prior methods.

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

# A   FOUNDATIONS

Here, we summarize the foundations of our online meta-learning formulation. In particular, we concentrate on model agnostic meta-learning and online (i.e. regret based) learning. To illustrate the differences in setting and algorithms, we will use the running example of few-shot learning, which we describe below first. We emphasize that online learning, MAML, and the online meta-learning formulations have a broader scope than few-shot supervised learning. We use the few-shot supervised learning example primarily for illustration.

## A.1   FEW-SHOT LEARNING

In the few-shot supervised learning setting (Santoro et al., 2016), we are interested in a family of tasks, where each task $\mathcal{T}$ is associated with a notional and infinite-size population of input-output pairs. In the few-shot learning, the goal is to learn a task while accessing only a small, finite-size labeled dataset $\mathcal{D}_i := \{\mathbf{x}_i, \mathbf{y}_i\}$ corresponding to task $\mathcal{T}_i$. If we have a predictive model, $\boldsymbol{h}(\cdot; \mathbf{w})$, with parameters $\mathbf{w}$, the population risk of the model is $f_i(\mathbf{w}) := \mathbb{E}_{(\mathbf{x},\mathbf{y}) \sim \mathcal{T}_i}[\ell(\mathbf{x}, \mathbf{y}, \mathbf{w})]$, where the expectation is over the task population and $\ell(\cdot)$ is a loss function, such as the square loss or cross-entropy between the model prediction and the correct label. For example, in th e case of square loss we have, $\ell(\mathbf{x}, \mathbf{y}, \mathbf{w}) = ||\mathbf{y} - \boldsymbol{h}(\mathbf{x}; \mathbf{w})||^2$. Let $\mathcal{L}(\mathcal{D}_i, \mathbf{w})$ represent the average loss on the dataset $\mathcal{D}_i$.

Being able to effectively minimize $f_i(\mathbf{w})$ is likely hard if we rely only on $\mathcal{D}_i$ due to the small size of the dataset. However, we are exposed to many such tasks from the family — either in sequence or as a batch, depending on the setting. By being able to draw upon the multiplicity of tasks, we may hope to perform better, as for example demonstrated in the meta-learning literature.

## A.2   META-LEARNING AND MAML

Meta-learning, or learning to learn (Schmidhuber, 1987), aims to effectively bootstrap from a set of tasks to learn faster on a new task. It is assumed that tasks are drawn from a fixed distribution, $\mathcal{T} \sim \mathbb{P}(\mathcal{T})$. At meta-training time, $M$ tasks $\{\mathcal{T}_i\}_{i=1}^M$ are drawn from this distribution and datasets corresponding to them are made available to the agent. At deployment time, we are faced with a new test task $\mathcal{T}_j \sim \mathbb{P}(\mathcal{T})$, for which we are again presented with a small labeled dataset $\mathcal{D}_j := \{\mathbf{x}_j, \mathbf{y}_j\}$. Meta-learning algorithms attempt to find a model using the $M$ training tasks, such that when $\mathcal{D}_j$ is revealed from the test task, the model can be quickly updated to minimize $f_j(\mathbf{w})$. MAML (Finn et al., 2017) does this by learning an initial set of parameters $\mathbf{w}_{\mathrm{MAML}}$, such that at meta-test time, performing a few steps of gradient descent from $\mathbf{w}_{\mathrm{MAML}}$ using $\mathcal{D}_j$ minimizes $f_j(\cdot)$. To get such an initialization, at meta-training time, MAML solves the optimization problem:

$$\mathbf{w}_{\mathrm{MAML}} := \arg\min_{\mathbf{w}} \ \frac{1}{M} \sum_{i=1}^{M} f_i\big(\mathbf{w} - \alpha \nabla \hat{f}_i(\mathbf{w})\big). \tag{4}$$

The inner gradient, $\nabla \hat{f}_i(\mathbf{w})$, is based on a mini-batch of data from $\mathcal{D}_i$. Hence, MAML optimizes for few-shot generalization. Note that the optimization problem is subtle: we have a gradient descent step embedded in the actual objective function. Regardless, Finn et al. (2017) show that gradient-based methods can be used on this optimization objective with existing automatic differentiation libraries. Stochastic optimization techniques are used to solve the optimization problem in Eq. (4) since the population risk is not known directly. At meta-test time, the solution to Eq. 4 is fine-tuned as: $\mathbf{w}_j \leftarrow \mathbf{w}_{\mathrm{MAML}} - \alpha \nabla \hat{f}_j(\mathbf{w}_{\mathrm{MAML}})$ with the gradient obtained using $\mathcal{D}_j$.

MAML and other meta-learning algorithms are not directly applicable to sequential settings for two reasons. First, they have two distinct phases: meta-training and meta-testing or deployment. We would like the algorithms to work in a continuous learning fashion. Second, meta-learning methods generally assume that the tasks come from some fixed distribution, whereas we would like methods that work for non-stationary task distributions.

## A.3   ONLINE LEARNING

In the online learning setting, an agent faces a sequence of loss functions $\{f_t\}_{t=1}^\infty$, one in each round $t$. These functions need not be drawn from a fixed distribution, and could even be chosen adversarially over time. The goal for the learner is to sequentially decide on model parameters $\{\mathbf{w}_t\}_{t=1}^\infty$ that

perform well on the loss sequence. In particular, the standard objective is to minimize some notion of regret defined as the difference between our learner's loss, $\sum_{t=1}^{T} f_t(\mathbf{w}_t)$, and the best performance achievable by some family of methods (comparator class). The most standard notion of regret is to compare to the cumulative loss of the best *fixed* model in hindsight:

$$\text{Regret}_T = \sum_{t=1}^{T} f_t(\mathbf{w}_t) - \min_{\mathbf{w}} \sum_{t=1}^{T} f_t(\mathbf{w}). \tag{5}$$

The goal in online learning is to design algorithms such that this regret grows with $T$ as slowly as possible. In particular, an agent (algorithm) whose regret grows sub-linearly in $T$ is non-trivially learning and adapting. One of the simplest algorithms in this setting is follow the leader (FTL) Hannan (1957), which updates the parameters as:

$$\mathbf{w}_{t+1} = \arg\min_{\mathbf{w}} \sum_{k=1}^{t} f_k(\mathbf{w}).$$

FTL enjoys strong performance guarantees depending on the properties of the loss function, and some variants use additional regularization to improve stability Shalev-Shwartz (2012). For the few-shot supervised learning example, FTL would consolidate all the data from the prior stream of tasks into a single large dataset and fit a single model to this dataset. As observed in the meta-learning literature, such a "joint training" approach may not learn effective models. To overcome this issue, we may desire a more adaptive notion of a comparator class, and algorithms that have low regret against such a comparator, as done in the online meta-learning formulation.

# B ANALYSIS AND PROOFS

In this section, outline the assumptions and proofs.

## B.1 ASSUMPTIONS

We make the following assumptions about each loss function $\{f_t, \hat{f}_t\}$ $\forall t$ in the learning problem. Let $\boldsymbol{\theta}$ and $\boldsymbol{\phi}$ represent two arbitrary choices of *model parameters*.

**Assumption 1.** *($C^2$-smoothness)*
*1. (Lipschitz in function value) $f$ has gradients bounded by $G$, i.e. $||\nabla f(\boldsymbol{\theta})|| \leq G \,\forall\, \boldsymbol{\theta}$.*

*2. (Lipschitz gradient) $f$ is $\beta-$smooth, i.e. $||\nabla f(\boldsymbol{\theta}) - \nabla f(\boldsymbol{\phi})|| \leq \beta ||\boldsymbol{\theta} - \boldsymbol{\phi}|| \,\forall (\boldsymbol{\theta}, \boldsymbol{\phi})$.*

*3. (Lipschitz Hessian) $f$ has $\rho-$Lipschitz Hessians, i.e. $||\nabla^2 f(\boldsymbol{\theta}) - \nabla^2 f(\boldsymbol{\phi})|| \leq \rho ||\boldsymbol{\theta} - \boldsymbol{\phi}|| \,\forall (\boldsymbol{\theta}, \boldsymbol{\phi})$.*

**Assumption 2.** *(Strong convexity) Suppose that $f$ is convex. Furthermore, suppose $f$ is $\mu-$strongly convex, i.e. $||\nabla f(\boldsymbol{\theta}) - \nabla f(\boldsymbol{\phi})|| \geq \mu ||\boldsymbol{\theta} - \boldsymbol{\phi}||$.*

These assumptions are largely standard in online learning, in various settings Cesa-Bianchi & Lugosi (2006), except 1.3. Examples where these assumptions hold include logistic regression and $L2$ regression over a bounded domain. Assumption 1.3 is a statement about the higher order smoothness of functions which is common in non-convex analysis Nesterov & Polyak (2006); Jin et al. (2017). In our setting, it allows us to characterize the landscape of the MAML-like function which has a gradient update step embedded within it. Importantly, these assumptions *do not* trivialize the meta-learning setting. A clear difference in performance between meta-learning and joint training can be observed even in the case where $f(\cdot)$ are quadratic functions, which correspond to the simplest strongly convex setting. See Appendix E for an example illustration.

## B.2 ANALYSIS

We analyze the FTML algorithm when the update procedure is a single step of gradient descent, as in the formulation of MAML. Concretely, the update procedure we consider is $\boldsymbol{U}_t(\mathbf{w}) = \mathbf{w} - \alpha \nabla \hat{f}_t(\mathbf{w})$. We restate our main theorem below for completeness.

**Theorem.** *Suppose $f$ and $\hat{f} : \mathbb{R}^d \to \mathbb{R}$ satisfy assumptions 1 and 2. Let $\tilde{f}$ be the function evaluated after a one step gradient update procedure, i.e.*

$$\tilde{f}(\mathbf{w}) := f\big(\mathbf{w} - \alpha \nabla \hat{f}(\mathbf{w})\big).$$

*If the step size is selected as $\alpha \leq \min\{\frac{1}{2\beta}, \frac{\mu}{8\rho G}\}$, then $\tilde{f}$ is convex. Furthermore, it is also $\tilde{\beta} = 9\beta/8$ smooth and $\tilde{\mu} = \mu/8$ strongly convex.*

*Proof.* First, the smoothness and strong convexity of $f$ and $\hat{f}$ implies $\mu \leq ||\nabla^2 \hat{f}(\boldsymbol{\theta})|| \leq \beta \,\forall\boldsymbol{\theta}$. Thus,

$$(1 - \alpha\beta) \leq ||\boldsymbol{I} - \alpha \nabla^2 \hat{f}(\boldsymbol{\theta})|| \leq (1 - \alpha\mu) \;\forall\boldsymbol{\theta}.$$

Also recall the earlier notation $\tilde{\boldsymbol{\theta}} = \boldsymbol{U}(\boldsymbol{\theta}) = \boldsymbol{\theta} - \alpha \nabla \hat{f}(\boldsymbol{\theta})$. For $\alpha < 1/\beta$, we have the following bounds:

$$
\begin{aligned}
(1 - \alpha\beta)||\boldsymbol{\theta} - \boldsymbol{\phi}|| \leq ||\boldsymbol{U}(\boldsymbol{\theta}) - \boldsymbol{U}(\boldsymbol{\phi})|| && \forall(\boldsymbol{\theta}, \boldsymbol{\phi}) \\
||\boldsymbol{U}(\boldsymbol{\theta}) - \boldsymbol{U}(\boldsymbol{\phi})|| \leq (1 - \alpha\mu)||\boldsymbol{\theta} - \boldsymbol{\phi}|| && \forall(\boldsymbol{\theta}, \boldsymbol{\phi}),
\end{aligned}
$$

since we have $\boldsymbol{U}(\boldsymbol{\theta}) - \boldsymbol{U}(\boldsymbol{\phi}) = \big(\boldsymbol{I} - \alpha \nabla^2 \hat{f}(\boldsymbol{\psi})\big)(\boldsymbol{\theta} - \boldsymbol{\phi})$ for some $\boldsymbol{\psi}$ that connects $\boldsymbol{\theta}$ and $\boldsymbol{\phi}$ due to the mean value theorem on $\nabla \hat{f}$. Using the chain rule and our definitions,

$$
\begin{aligned}
\nabla \tilde{f}(\boldsymbol{\theta}) - \nabla \tilde{f}(\boldsymbol{\phi}) &= \nabla \boldsymbol{U}(\boldsymbol{\theta}) \nabla f(\tilde{\boldsymbol{\theta}}) - \nabla \boldsymbol{U}(\boldsymbol{\phi}) \nabla f(\tilde{\boldsymbol{\phi}}) \\
&= \big(\nabla \boldsymbol{U}(\boldsymbol{\theta}) - \nabla \boldsymbol{U}(\boldsymbol{\phi})\big) \nabla f(\tilde{\boldsymbol{\theta}}) + \nabla \boldsymbol{U}(\boldsymbol{\phi}) \big(\nabla f(\tilde{\boldsymbol{\theta}}) - \nabla f(\tilde{\boldsymbol{\phi}})\big)
\end{aligned}
$$

Taking the norm on both sides, for the specified $\alpha$, we have:

$$||\nabla \tilde{f}(\boldsymbol{\theta}) - \nabla \tilde{f}(\boldsymbol{\phi})|| \leq || \left(\nabla \boldsymbol{U}(\boldsymbol{\theta}) - \nabla \boldsymbol{U}(\boldsymbol{\phi})\right) \nabla f(\tilde{\boldsymbol{\theta}})||$$
$$+ ||\nabla \boldsymbol{U}(\boldsymbol{\phi}) \left(\nabla f(\tilde{\boldsymbol{\theta}}) - \nabla f(\tilde{\boldsymbol{\phi}})\right) ||$$
$$\leq (\alpha \rho G + (1 - \alpha\mu)^2 \beta)||\boldsymbol{\theta} - \boldsymbol{\phi}||$$
$$\leq \left(\frac{\mu}{8} + \beta\right)||\boldsymbol{\theta} - \boldsymbol{\phi}||$$
$$\leq \frac{9\beta}{8}||\boldsymbol{\theta} - \boldsymbol{\phi}||.$$

Similarly, we obtain the following lower bound

$$||\nabla \tilde{f}(\boldsymbol{\theta}) - \nabla \tilde{f}(\boldsymbol{\phi})|| \geq ||\nabla \boldsymbol{U}(\boldsymbol{\phi}) \left(\nabla f(\tilde{\boldsymbol{\theta}}) - \nabla f(\tilde{\boldsymbol{\phi}})\right) ||$$
$$- || \left(\nabla \boldsymbol{U}(\boldsymbol{\theta}) - \nabla \boldsymbol{U}(\boldsymbol{\phi})\right) \nabla f(\tilde{\boldsymbol{\theta}})||$$
$$\geq (1 - \alpha\beta)^2 \mu||\boldsymbol{\theta} - \boldsymbol{\phi}|| - \alpha \rho G||\boldsymbol{\theta} - \boldsymbol{\phi}||$$
$$\geq \left(\frac{\mu}{4} - \frac{\mu}{8}\right)||\boldsymbol{\theta} - \boldsymbol{\phi}||$$
$$\geq \frac{\mu}{8}||\boldsymbol{\theta} - \boldsymbol{\phi}||$$

which completes the proof. $\square$

The following corollary is now immediate.

**Corollary.** *(inherited convexity for the MAML objective) If $\{f_i, \hat{f}_i\}_{i=1}^K$ satisfy assumptions 1 and 2, then the MAML optimization problem,*

$$\underset{\mathbf{w}}{minimize} \ \frac{1}{M} \sum_{i=1}^{M} f_i\big(\mathbf{w} - \alpha \nabla \hat{f}_i(\mathbf{w})\big),$$

*with $\alpha \leq \min\{\frac{1}{2\beta}, \frac{\mu}{8\rho G}\}$ is convex. Furthermore, it is $9\beta/8$-smooth and $\mu/8$-strongly convex.*

Since the objective function is convex, we may expect first-order optimization methods to be effective, since gradients can be efficiently computed with standard automatic differentiation libraries (as discussed in Finn et al. (2017)). In fact, this work provides the first set of results (under any assumptions) under which MAML-like objective function can be provably and efficiently optimized.

Another immediate corollary of our main theorem is that FTML now enjoys the same regret guarantees (up to constant factors) as FTL does in the comparable setting (with strongly convex losses).

**Corollary.** *(inherited regret bound for FTML) Suppose that for all $t$, $f_t$ and $\hat{f}_t$ satisfy assumptions 1 and 2. Suppose that the update procedure in FTML (Eq. 2) is chosen as $\boldsymbol{U}_t(\mathbf{w}) = \mathbf{w} - \alpha \nabla \hat{f}_t(\mathbf{w})$ with $\alpha \leq \min\{\frac{1}{2\beta}, \frac{\mu}{8\rho G}\}$. Then, FTML enjoys the following regret guarantee*

$$\sum_{t=1}^{T} f_t\big(\boldsymbol{U}_t(\mathbf{w}_t)\big) - \min_{\mathbf{w}} \sum_{t=1}^{T} f_t\big(\boldsymbol{U}_t(\mathbf{w})\big) = \mathcal{O}\left(\frac{32G^2}{\mu} \log T\right)$$

*Proof.* From Theorem 1, we have that each function $\tilde{f}_t(\mathbf{w}) = f_t(\boldsymbol{U}_t(\mathbf{w}))$ is $\tilde{\mu} = \mu/8$ strongly convex. The FTML algorithm is identical to FTL on the sequence of loss functions $\{\tilde{f}_t\}_{t=1}^T$, which has a $\mathcal{O}(\frac{4G^2}{\tilde{\mu}} \log T)$ regret guarantee (see Cesa-Bianchi & Lugosi (2006) Theorem 3.1). Using $\tilde{\mu} = \mu/8$ completes the proof. $\square$

More generally, our main theorem implies that there exists a large family of online meta-learning algorithms that enjoy sub-linear regret, based on the inherited smoothness and strong convexity of $\tilde{f}(\cdot)$. See Hazan (2016); Shalev-Shwartz (2012); Shalev-Shwartz & Kakade (2008) for algorithmic templates to derive sub-linear regret based algorithms.

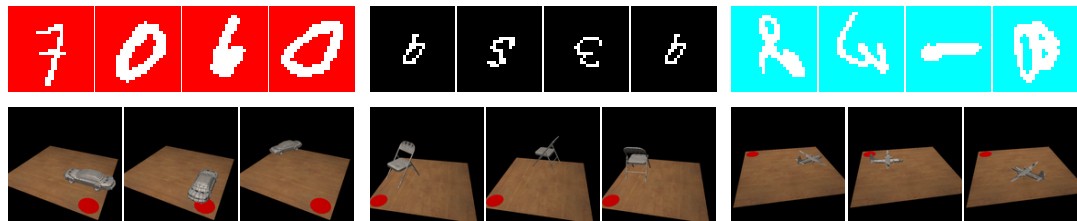

Figure 2: Illustration of three tasks for Rainbow MNIST (top) and pose prediction (bottom). CIFAR images not shown. Rainbow MNIST includes different rotations, scaling factors, and background colors. For the pose prediction tasks, the goal is to predict the global position and orientation of the object on the table. Cross-task variation includes varying 50 different object models within 9 object classes, varying object scales, and different camera viewpoints.

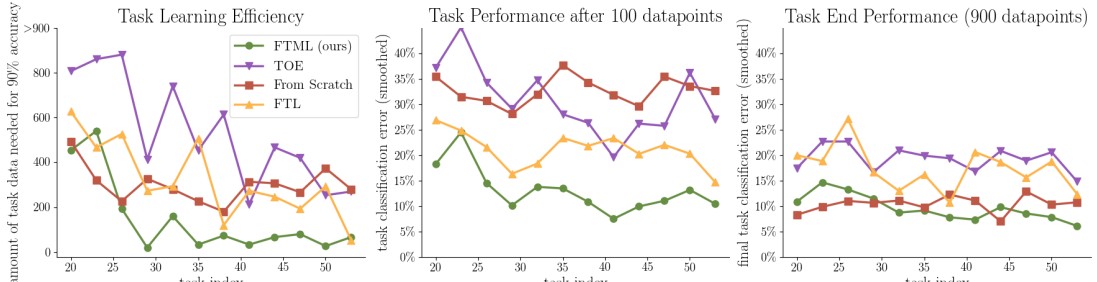

Figure 3: Rainbow MNIST results. Left: amount of data needed to learn each new task. Center: task performance after 100 datapoints on the current task. Right: The task performance after all 900 datapoints for the current task have been received. Lower is better for all plots. FTML can learn new tasks more and more efficiently as each new task is received, demonstrating effective forward transfer.

## C  EXTENDED EXPERIMENTAL EVALUATION

In this appendix, we present the extended experimental evaluation that is introduced in Section 3.

### C.1  RAINBOW MNIST

In this experiment, we create a sequence of tasks based on the MNIST character recognition dataset. We transform the digits in a number of ways to create different tasks, such as 7 different colored backgrounds, 2 scales (half size and original size), and 4 rotations of 90 degree intervals. As illustrated in Figure 2, a task involves correctly classifying digits with a randomly sampled background, scale, and rotation. This leads to 56 total tasks. We partitioned the MNIST training dataset into 56 batches of examples, each with 900 images and applied the corresponding task transformation to each batch of images. The ordering of tasks was selected at random and we set $90\%$ classification accuracy as the proficiency threshold.

The learning curves in Figure 3 show that FTML learns tasks more and more quickly, with each new task added. We also observe that FTML substantially outperforms the alternative approaches in both efficiency and final performance. FTL performance better than TOE since it performs task-specific adaptation, but its performance is still inferior to FTML. We hypothesize that, while the prior methods improve in efficiency over the course of learning as they see more tasks, they struggle to prevent negative transfer on each new task. Our last observation is that training independent models does not learn efficiently, compared to models that incorporate data from other tasks; but, their final performance with 900 data points is similar.

### C.2  FIVE-WAY CIFAR-100

In this experiment, we create a sequence of 5-way classification tasks based on the CIFAR-100 dataset, which contains more challenging and realistic RGB images than MNIST. Each classification problem involves a newly-introduced class from the 100 classes in CIFAR-100. Thus, different tasks correspond to different labels spaces. The ordering of tasks is selected at random, and we measure performance using classification accuracy. Since it is less clear what the proficiency threshold should

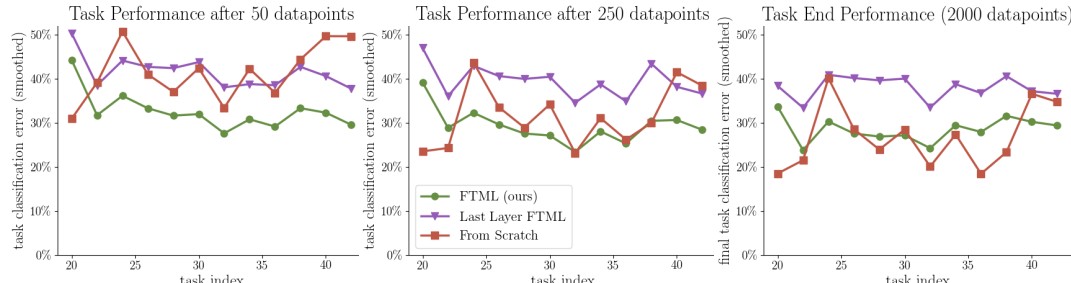

Figure 4: Online CIFAR-100 results, evaluating task performance after 50, 250, and 2000 datapoints have been received for a given task. We see that FTML learns each task much more efficiently than models trained from scratch, while both achieve similar asymptotic performance after 2000 datapoints. We also observe that FTML benefits from adapting all layers rather than learning a shared feature space across tasks while adapting only the last layer.

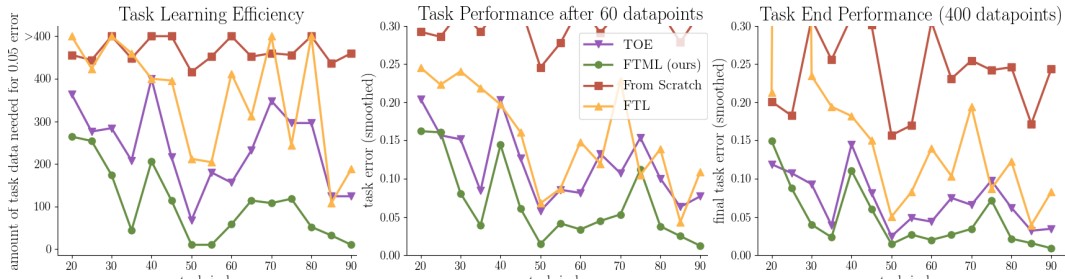

Figure 5: Object pose prediction results. On the left, we observe that online meta-learning generally leads to faster learning as more and more tasks are introduced, learning with only 10 datapoints for many of the tasks. In the center and right, we see that meta-learning enables transfer not just for faster learning but also for more effective performance when 60 and 400 datapoints of each task are available. Note that the order of tasks is randomized, hence leading to spikes when more difficult tasks are introduced.

be for this task, we evaluate the accuracy on each task after varying numbers of datapoints have been seen. Since these tasks are mutually exclusive (as label space is changing), it makes sense to train the TOE model with a different final layer for each task. An extremely similar approach to this is to use our meta-learning approach but to only allow the final layer parameters to be adapted to each task. Further, such a meta-learning approach is a more direct comparison to our full FTML method, and the comparison can provide insight into whether online meta-learning is simply learning features and performing training on the last layer, or if it is adapting the features to each task. Thus, we compare to this last layer online meta-learning approach instead of TOE with multiple heads. The results (see Figure 4) indicate that FTML learns more efficiently than independent models and a model with a shared feature space. The results on the right indicate that training from scratch achieves good performance with 2000 datapoints, reaching similar performance to FTML. However, the last layer variant of FTML seems to not have the capacity to reach good performance on all tasks.

## C.3 SEQUENTIAL OBJECT POSE PREDICTION

In our final experiment, we study a 3D pose prediction problem. Each task involves learning to predict the global position and orientation of an object in an image. We construct a dataset of synthetic images using 50 object models from 9 different object classes in the PASCAL3D+ dataset (Xiang et al., 2014), rendering the objects on a table using the renderer accompanying the MuJoCo physics engine (Todorov et al., 2012) (see Figure 2). To place an object on the table, we select a random 2D location, as well as a random azimuthal angle. Each task corresponds to a different object with a randomly sampled camera angle. We place a red dot on one corner of the table to provide a global reference point for the position. Using this setup, we construct 90 tasks (with an average of about 2 camera viewpoints per object), with 1000 datapoints per task. All models are trained to regress to the global 2D position and the sine and cosine of the azimuthal angle (the angle of rotation along the z-axis). For the loss functions, we use mean-squared error, and set the proficiency threshold to an error of 0.05. We show the results of this experiment in Figure 5. The results demonstrate that

meta-learning can improve both efficiency and performance of new tasks over the course of learning, solving many of the tasks with only 10 datapoints. Unlike the previous settings, TOE substantially outperforms training from scratch, indicating that it can effectively make use of the previous data from other tasks, likely due to the greater structural similarity between the pose detection tasks. However, the performance of FTML suggests that even better transfer can be accomplished by explicitly optimizing for the ability to quickly and effectively learn new tasks. Finally, we find that FTL performs comparably or worse than TOE, indicating that task-specific fine-tuning can lead to overfitting when the model is not explicitly trained for the ability to fine-tune effectively.

## D  CONNECTIONS TO RELATED WORK

Our work proposes to use meta-learning or learning to learn Thrun & Pratt (1998); Schmidhuber (1987); Naik & Mammone (1992), in the context of online (regret-based) learning. We reviewed the foundations of these approaches in Section A, and we summarize additional related work along different axis.

**Meta-learning:** Prior meta-learning works have proposed learning an update rule or optimizer for fast adaptation Hochreiter et al. (2001); Bengio et al. (1992); Andrychowicz et al. (2016); Li & Malik (2017); Ravi & Larochelle (2017) by having one model output the weights of another Ha et al. (2017), while other works have used recurrent models that learn by ingesting datasets directly Santoro et al. (2016); Duan et al. (2016); Wang et al. (2016); Munkhdalai & Yu (2017); Mishra et al. (2017). While some meta-learning works have considered online learning settings at *meta-test time* Santoro et al. (2016); Al-Shedivat et al. (2017); Nagabandi et al. (2018), nearly all prior meta-learning algorithms assume that the *meta-training tasks* come from a stationary distribution. Furthermore, most prior work has not evaluated versions of meta-learning algorithms when presented with a continuous stream of tasks. Recent work has considered handling non-stationary task distributions in meta-learning using Dirichlet process mixture models over meta-learned parameters Grant et al. (2019). Unlike this prior work, we introduce a simple extension onto the MAML algorithm without mixtures over parameters, and provide theoretical guarantees.

**Continual learning:** Our problem setting is related to (but distinct from) continual, or lifelong learning Thrun (1998); Zhao & Schmidhuber (1996). In lifelong learning, a number of recent papers have focused on avoiding forgetting and negative backward transfer Goodfellow et al. (2013); Kirkpatrick et al. (2017); Zenke et al. (2017); Rebuffi et al. (2017); Shin et al. (2017); Shmelkov et al. (2017); Lopez-Paz et al. (2017); Nguyen et al. (2017). Other papers have focused on maintaining a reasonable model capacity as new tasks are added Lee et al. (2017); Mallya & Lazebnik (2017). In this paper, we sidestep the problem of catastrophic forgetting by maintaining a buffer of all the observed data Isele & Cosgun (2018). In future work, we hope to understand the interplay between limited memory and catastrophic forgetting for variants of the FTML algorithm. Here, we instead focuses on the problem of forward transfer – maximizing the efficiency of learning new tasks within a non-stationary learning setting. Prior works have also considered settings that combine joint training across tasks with task-specific adaptation Barto et al. (1995); Lowrey et al. (2019), but have not explicitly employed meta-learning. Furthermore, unlike prior works Ruvolo & Eaton (2013); Rusu et al. (2016); Aljundi et al. (2017); Wang et al. (2017), we also focus on the setting where there are several tens or hundreds of tasks. This setting is interesting since there is significantly more information that can be transferred from previous tasks and we can employ more sophisticated techniques such as meta-learning for transfer, enabling the agent to move towards few-shot learning after experiencing a large number of tasks.

**Online learning:** Similar to continual learning, online learning deals with a sequential setting with streaming tasks. It is well known in online learning that FTL has good regret guarantees, but is often computationally expensive. Thus, there is a large body of work on developing computationally cheaper algorithms Cesa-Bianchi & Lugosi (2006); Hazan et al. (2006); Zinkevich (2003); Shalev-Shwartz (2012). Again, in this work, we sidestep the computational considerations to first study if the meta-learning analog of FTL can provide performance gains. For this, we derived the FTML algorithm which has low regret when compared to a powerful adaptive comparator class that performs task-specific adaptation. We leave the design of more computationally efficient versions of FTML to future work. To avoid the pitfalls associated with a single best model in hindsight, online learning literature has also studied alternate notions of regret, with the closest settings being dynamic regret and adaptive or tracking regret. In the dynamic regret setting (Herbster & Warmuth, 1995; Yang et al., 2016; Besbes et al., 2015), the performance of the online learner's model sequence is compared

against the sequence of optimal solutions corresponding to each loss function in the sequence. Unfortunately, lower-bounds (Yang et al., 2016) suggest that the comparator class is too powerful and may not provide for any non-trivial learning in the general case. To overcome these limitations, prior work has placed restrictions on how quickly the loss functions or the comparator model can change (Hazan & Comandur, 2009; Hall & Willett, 2015; Herbster & Warmuth, 1995). In contrast, we consider a different notion of adaptive regret, where the learner and comparator both have access to an update procedure. The update procedures allow the comparator to produce different models for different loss functions, thereby serving as a powerful comparator class (in comparison to a fixed model in hindsight). For this setting, we derived sublinear regret algorithms without placing any restrictions on the sequence of loss functions. We believe that this setting captures the spirit and practice of continual lifelong learning, and also leads to promising empirical results.

## E  LINEAR REGRESSION EXAMPLE

Here, we present a simple example of optimizing a collection of quadratic objectives (equivalent to linear regression on fixed set of features), where the solutions to joint training and the meta-learning (MAML) problem are different. The purpose of this example is to primarily illustrate that meta-learning can provide performance gains even in seemingly simple and restrictive settings. Consider a collection of objective functions: $\{f_i : \mathbf{w} \in \mathbb{R}^d \to \mathbb{R}\}_{i=1}^M$ which can be described by quadratic forms. Specifically, each of these functions are of then form

$$f_i(\mathbf{w}) = \frac{1}{2}\mathbf{w}^T \boldsymbol{A}_i \mathbf{w} + \mathbf{w}^T \boldsymbol{b}_i.$$

This can represent linear regression problems as follows: let $(\mathbf{x}_{\mathcal{T}_i}, \mathbf{y}_{\mathcal{T}_i})$ represent input-output pairs corresponding to task $\mathcal{T}_i$. Let the predictive model be $\boldsymbol{h}(\mathbf{x}) = \mathbf{w}^T\mathbf{x}$. Here, we assume that a constant scalar (say 1) is concatenated in $\mathbf{x}$ to subsume the constant offset term (as common in practice). Then, the loss function can be written as:

$$f_i(\mathbf{w}) = \frac{1}{2}\mathbb{E}_{(\mathbf{x},\mathbf{y})\sim\mathcal{T}_i}\left[||\boldsymbol{h}(\mathbf{x}) - \mathbf{y}||^2\right]$$

which corresponds to having $\boldsymbol{A}_i = \mathbb{E}_{\mathbf{x}\sim\mathcal{T}_i}[\mathbf{x}\mathbf{x}^T]$ and $\boldsymbol{b}_i = \mathbb{E}_{(\mathbf{x},\mathbf{y})\sim\mathcal{T}_i}[\mathbf{x}^T\mathbf{y}]$. For these set of problems, we are interested in studying the difference between joint training and meta-learning.

**Joint training**  The first approach of interest is joint training which corresponds to the optimization problem

$$\min_{\mathbf{w}\in\mathbb{R}^d} F(\mathbf{w}), \text{ where } F(\mathbf{w}) = \frac{1}{M}\sum_{i=1}^M f_i(\mathbf{w}). \tag{6}$$

Using the form of $f_i$, we have

$$F(\mathbf{w}) = \frac{1}{2}\mathbf{w}^T\left(\frac{1}{M}\sum_{i=1}^M \boldsymbol{A}_i\right)\mathbf{w} + \mathbf{w}^T\left(\frac{1}{M}\sum_{i=1}^M \boldsymbol{b}_i\right).$$

Let us define the following:

$$\bar{\boldsymbol{A}} := \frac{1}{M}\sum_{i=1}^M \boldsymbol{A}_i \text{ and } \bar{\boldsymbol{b}} := \frac{1}{M}\sum_{i=1}^M \boldsymbol{b}_i.$$

The solution to the joint training optimization problem (Eq. 6) is then given by $\mathbf{w}_{\text{joint}}^* = -\bar{\boldsymbol{A}}^{-1}\bar{\boldsymbol{b}}$.

**Meta learning (MAML)**  The second approach of interest is meta-learning, which as mentioned in Section A.2 corresponds to the optimization problem:

$$\min_{\mathbf{w}\in\mathbb{R}^d} \tilde{F}(\mathbf{w}), \text{ where } \tilde{F}(\mathbf{w}) = \frac{1}{M}\sum_{i=1}^M f_i(\boldsymbol{U}_i(\mathbf{w})). \tag{7}$$

Here, we specifically concentrate on the 1-step (exact) gradient update procedure: $\boldsymbol{U}_i(\mathbf{w}) = \mathbf{w} - \alpha\nabla f_i(\mathbf{w})$. In the case of the quadratic objectives, this leads to:

$$f_i(\boldsymbol{U}_i(\mathbf{w})) = \frac{1}{2}(\mathbf{w} - \alpha\boldsymbol{A}_i\mathbf{w} - \alpha\boldsymbol{b}_i)^T \boldsymbol{A}_i(\mathbf{w} - \alpha\boldsymbol{A}_i\mathbf{w} - \alpha\boldsymbol{b}_i)$$
$$+ (\mathbf{w} - \alpha\boldsymbol{A}_i\mathbf{w} - \alpha\boldsymbol{b}_i)^T \boldsymbol{b}_i$$

The corresponding gradient can be written as:

$$\nabla f_i(\boldsymbol{U}_i(\mathbf{w})) = \left(\boldsymbol{I} - \alpha\boldsymbol{A}_i\right)\left(\boldsymbol{A}_i\left(\mathbf{w} - \alpha\boldsymbol{A}_i\mathbf{w} - \alpha\boldsymbol{b}_i\right) + \boldsymbol{b}_i\right)$$

$$= \left(\boldsymbol{I} - \alpha\boldsymbol{A}_i\right)\boldsymbol{A}_i\left(\boldsymbol{I} - \alpha\boldsymbol{A}_i\right)\mathbf{w} + \left(\boldsymbol{I} - \alpha\boldsymbol{A}_i\right)^2\boldsymbol{b}_i$$

For notational convenience, we define:

$$\boldsymbol{A}_\dagger := \frac{1}{M}\sum_{i=1}^{M}\left(\boldsymbol{I} - \alpha\boldsymbol{A}_i\right)^2\boldsymbol{A}_i$$

$$\boldsymbol{b}_\dagger := \frac{1}{M}\sum_{i=1}^{M}\left(\boldsymbol{I} - \alpha\boldsymbol{A}_i\right)^2\boldsymbol{b}_i.$$

Then, the solution to the MAML optimization problem (Eq. 7) is given by $\mathbf{w}^*_{\text{MAML}} = -\boldsymbol{A}_\dagger^{-1}\boldsymbol{b}_\dagger$.

**Remarks**   In general, $\mathbf{w}^*_{\text{joint}} \neq \mathbf{w}^*_{\text{MAML}}$ based on our analysis. Note that $\boldsymbol{A}_\dagger$ is a weighed average of different $\boldsymbol{A}_i$, but the weights themselves are a function of $\boldsymbol{A}_i$. The reason for the difference between $\mathbf{w}^*_{\text{joint}}$ and $\mathbf{w}^*_{\text{MAML}}$ is the difference in moments of input distributions. The two solutions, $\mathbf{w}^*_{\text{joint}}$ and $\mathbf{w}^*_{\text{MAML}}$, coincide when $\boldsymbol{A}_i = \boldsymbol{A}\ \forall i$. Furthermore, since $\mathbf{w}^*_{\text{MAML}}$ was optimized to explicitly minimize $\tilde{F}(\cdot)$, it would lead to better performance after task-specific adaptation.

This example and analysis reveals that there is a clear separation in performance between joint training and meta-learning even in the case of quadratic loss functions. Improved performance with meta-learning approaches have been noted empirically with non-convex loss landscapes induced by neural networks. Our example illustrates that meta learning can provide non-trivial gains over joint training even in simple convex loss landscapes.

