# OpenReview forum: "Online Meta-Learning"
_ICLR.cc/2019/Workshop/LLD — LLD 2019_

### Official Review · AnonReviewer3 · 2019-04-12
**A new paradigm for continual lifelong learning**

**Rating:** 4
**Confidence:** 2

**Review:**

This paper formulated a new learning paradigm that combines meta-learning and online leanring, which is more general than few-shot supervised learning paradigm. The authors proposed a FTL-fashioned algorithm (FTML) that extends MAML to the online setting. FTML achieves a regret of order O(logT) under some C^2-smoothness assumption and a \mu-strongly convex loss. This logarithmic regret bound is comparable to the usual FTL algorithms in a similar setting. The provable algorithm is however not straightforwardly applicable in practice, but it is shown that a MAML-typed modification can lead to reasonable performances.

Regarding the theoretical part, the contribution does not seem to be technically significant (I don't really have time to check the analysis so I may be wrong) but provided a first set of results to the new paradigm. One flaw is maybe that the implemented algorithm is not exactly the same as the provable one, but it is comprehensible...

The experimental setting, in particular the choice of baselines is reasonable and integrated since there are no real prior algorithms on this new paradigm. I somehow don't like the fact that no uncertainty are given in the figures (or at least the number of replications of each experiment could've been reported).

Overall, the paper may lack of some self-containedness due to the page limits, but remains a sound enough paper for the workshop.

I would vote for accept.

Minor comments:
1. The detailed assumptions have been placed in Appendix for the sake of space constraint, but in Corollary 1 it is stated that "under assumptions 1 and 2..." where no clue is given on where to find assumptions 1 and 2. Well I finally found them in the appendix, but it took me a bit of time.
2. I personally don't like a sole subsection indexed x.1 within a whole section, well it's a personal taste...

---

### Official Review · AnonReviewer2 · 2019-04-12
**The paper seems to be sound**

**Rating:** 4
**Confidence:** 2

**Review:**

The paper introduces a task of online metalearning, where the agent is doing few shot learning online -- every task is seen only once.

I am not an expert in online learning, but the paper seems to be sound. The experimentation looks thorough. The results are promising, but I would like to see some dataset closer to real world.

A minor point -- graphs are very hard to read, please use vector images

---

### Official Review · AnonReviewer4 · 2019-04-13
**Review of "Online Meta-Learning"**

**Rating:** 4
**Confidence:** 2

**Review:**

Summary of the paper:

This work proposes a “best of both worlds approach”, by introducint an online meta-learning algorithm.
The “follow the meta leader” algorithm (and its analysis) heavily builds on the “follow the leader” algorithm from online convex optimization, which leaves the door open for future improvements.
Some numerical experiments favorably comparing the approach with previous work are provided.

A few comments and questions:
-there is a (small) typo, line 7 of section A1 page 8, in the appendix
-second corollary, page 11: why put a 32 in the big O notation (same comment for the proof)?

Reviewer’s assessment:
I found the paper to be well written. The ideas are exposed clearly and the numerical results support the approach. Since the problem tackled by this work clearly falls within the scope of the workshop, I recommend to accept this paper.

---

### Decision · Program_Chairs · 2019-04-16
**Acceptance Decision**

Accept